# Chartalist: Labeled Graph Datasets for UTXO and Account-based Blockchains

**Kiarash Shamsi**
Computer Science
University of Manitoba
`shamsik1@myumanitoba.ca`

**Friedhelm Victor**
Computer Science
TU Berlin
`friedhelm.victor@tu-berlin.de`

**Murat Kantarcioglu**
Computer Science
UT Dallas
`muratk@utdallas.edu`

**Yulia R. Gel**
Statistics
UT Dallas
`ygl@utdallas.edu`

**Cuneyt G. Akcora**
Computer Science and Statistics
University of Manitoba
`cuneyt.akcora@umanitoba.ca`

## Abstract

Machine learning on blockchain graphs is an emerging field with many applications such as ransomware payment tracking, price manipulation analysis, and money laundering detection. However, analyzing blockchain data requires domain expertise and computational resources, which pose a significant barrier and hinder advancement in this field. We introduce Chartalist, the first comprehensive platform to methodically access and use machine learning across a large selection of blockchains to address this challenge.

Chartalist contains ML-ready datasets from unspent transaction output (UTXO) (e.g., Bitcoin) and account-based blockchains (e.g., Ethereum). We envision that Chartalist can facilitate data modeling, analysis, and representation of blockchain data and attract a wider community of scientists to analyze blockchains. Chartalist is a sustainable open-science initiative at `https://github.com/cakcora/Chartalist`.

## 1 Introduction

Blockchain is an emerging technology that has already enabled a wide range of applications, from cryptocurrencies (e.g., Bitcoin and Ethereum) and digital asset management to accountable data sharing. Most of the blockchain information can be represented as single and multilayer graphs composed of different types of nodes or edges. For example, for the Ethereum blockchain, the transactions that trade a digital token or asset can be seen to create a unique layer between the involved addresses (i.e., nodes of the transaction graph). Due to the increased popularity, analyzing the graph data stored on blockchains has emerged as an essential data science and machine learning (ML) problem. For example, building ML models (e.g., graph neural networks) using cryptocurrency blockchain graphs has direct applications for ransomware payment tracking, price manipulation analysis, and money laundering detection.

Analyzing the massive blockchain data today requires significant efforts to extract the underlying graph by running a Bitcoin client or paying for commercial APIs (e.g., etherscan.io) to download transaction data. Even when we extract transaction data by any means, we must allocate considerable resources to construct blockchain graphs. For example, a leading open-source Bitcoin parser BlockSci [19] requires 60GB of memory to build the Bitcoin transaction graph. Ethereum poses even greater challenges as its data is bigger and requires smart contract analysis to discover internal transactions. Combined with data size and complexity issues, blockchain research also lacks labeled

data for many significant problems – these issues plague ML on blockchains and force researchers to develop their pipelines to work with blockchain data.

To enable broader ML research in this emerging area, we created a dataset repository named Chartalist, containing cleaned and labeled data (e.g., known ransomware payment addresses) combined with open-source data loaders and graph extractors for easy analysis and model building using the provided data. To the best of our knowledge, Chartalist is the first attempt to systematically organize blockchain data for the broader ML community and provide a set of ML tasks defined for appropriate blockchain datasets.

In addition to facilitating ML applications on blockchains, our dataset further enables ML model development for dynamic multilayer graphs. In particular, although there are many publicly available multilayer graph datasets, the existing graphs are either small, static, focus on a specific case when the set of nodes is fixed across layers (i.e., multiplex networks), or lack ground truth information. To our knowledge, *no dynamic multilayer graphs with ground truth for anomaly detection are publicly available*. In contrast, our dataset provides large-scale, dynamic multilayer networks where nodes, edges, and edge weights evolve. Furthermore, adopting and curating major blockchain events allow us to complement such multilayer graphs with the ground truth information on event anomalies. The graph data given in Chartalist can be used for a broad range of ML tasks, such as forecasting, link prediction, and node classification with graph neural networks (GNNs).

**Sustainability of Chartalist.** Data Security and Privacy Lab of UT Dallas and Future Data Lab of the UManitoba are committed to hosting, managing, and updating the Chartalist Data Repository at least quarterly.

Chartalist contributes to graph ML research in the multifold ways:

- Chartalist removes the burden of downloading, parsing, and cleaning blockchain data, the key bottleneck for analysis of blockchain data by the broader ML community. Chartalist aims to be a reliable supplier of Bitcoin and Ethereum blockchain graphs and plays a vital role in emerging graph machine learning.

- Chartalist intends to be the focal point for curated address and edge labels that are collected from the community.

- Chartalist provides transaction and block extraction and graph creation tools to facilitate graph machine learning on blockchains.

## 2   Related Work

**Relation to Data Repositories.**  A few repositories contain graph datasets with information on the node and edge labels, along with graph labels and tasks. For example, the Stanford Network Repository (SNAP) curates various graph types under umbrella terms, such as transportation networks, social networks, and web networks. TUDatasets [26] curates benchmark datasets for evaluating graph kernels and graph neural networks. NetworkDataRepository [33] is an all-purpose graph repository that also provides visual and detailed descriptive statistics on graphs. While these repositories are essential for data sharing and curation, they do not contain ML-ready blockchain datasets (e.g., well-annotated metadata, temporal data, multi-layered networks, label information). There are a few reasons for this exclusion. First, blockchain graphs are dynamic and require extensive domain expertise to extract transaction data, process the network, and construct datasets that can be used for machine learning. Second, repositories may shun blockchain datasets due to large transaction network sizes (the Bitcoin blockchain is well over 350GB in storage). For example, SNAP only contains a Bitcoin trust dataset from a trading site unrelated to the transaction network. In addition, existing data repositories do not contain time-evolving multilayer graphs with ground truth information suitable for benchmarking anomaly detection methods.

**Relation to ML Benchmarks.** Benchmarks are critical in tracking state of the art in ML (e.g., Open Graph Benchmark [16]). To our knowledge, there is no benchmark for graph machine learning on blockchain data. However, single labeled blockchain datasets such as [39] are now widely used in graph neural networks, and performance results are compared in research papers. More related to us, PapersWithCode [33] provides state-of-the-art results on various graph tasks, such as link prediction

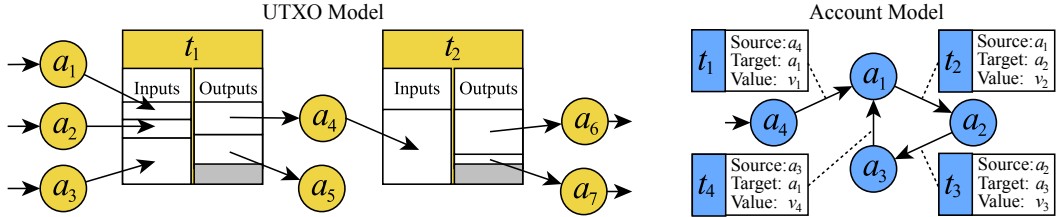

Figure 1: In the UTXO model, transactions (i.e. $t_1, t_2$) between addresses ($a_i$) can consist of multiple inputs and outputs. In the account model, every transaction has only one source and target address.

and graph classification. However, the website has not defined a blockchain task yet. We aim to fill the blockchain benchmark role with Chartalist.

**Relation to Blockchain Analytic Tools.** A few open-source tools exist for graph machine learning on blockchains. Notably, BlockSci [19] implements address clustering for UTXO-based blockchains. Similarly, Blockchain-ETL is an open-source framework for extracting transaction data from both UTXO and account-based blockchains; Egret [32] provides a coin tracking and transaction visualization tool for Bitcoin. However, these tools do not store and share blockchain graphs. Analytics companies such as chainalysis.com and etherscan.io provide online tracking and visualization tools for most blockchains; however, their products are not open-sourced. For an overview of tools and resources, visit the Chartalist website.

## 3 Overview of Chartalist Data Commons

Chartalist has three main components: 1) A holistic view of blockchains and formulations of graph machine learning tasks. 2) A comprehensive ecosystem of tools and community resources to support blockchain data analytics. 3) A set of boards to support performance comparison and benchmark for the tasks. As a first step, we now provide an overview of blockchain types that shape our approach in ML model building using the blockchain transaction network.

### 3.1 Blockchain Data Types

Due to the numerous advantages of blockchain technology, blockchain-based platforms are growing very fast. Although the fundamental concepts behind all these networks are very similar, They may use various architectures with different data types to implement their network. For instance, Bitcoin (2008) [27] is the first implementation of blockchain technology where coin transaction are the primary data to be stored; However more recent platforms such as Ethereum has been created to store more complicated structures like software code, called smart contracts.

Blockchains can be categorized into two broad lines in terms of transaction type: unspent transaction output (UTXO)-based (e.g., Bitcoin, Litecoin) and account-based (e.g., Ethereum) blockchains [7]. The difference between UTXO and account-based models profoundly impacts blockchain networks [41]. Figure 1 illustrates UTXO and account model transaction networks. An analogy to account-based blockchains is bank accounts that can be used to make payments and keep a remaining balance. In these blockchains, a transaction has exactly one input and one output address (an address is a unique account identifier on the blockchain transaction network). An address may be used to receive and send coins multiple times. The resulting network is similar to traditional social networks, which implies that social network analysis tools can be directly applied to account networks. Smart contracts are a more recent concept mainly available on account-based platforms [44]. These contacts have their own account addresses which can be called to enable certain actions such as buying/selling digital tokens [45]. An account-based smart contract platform employs code accounts to manage application-specific states, such as asset balances. These state changes can result in a diverse set of asset networks. The networks may share the same set of account addresses, allowing us to link activity, observe asset flows in the network, and create multilayer networks (i.e., flows associated with each asset result in a different layer).

In turn, the second type (i.e., UTXO-based blockchains), such as Bitcoin, have made design choices to blur the association between input and output addresses of a transaction [5]. First, transactions may

involve more than two addresses. Users often use many addresses, and it is not straightforward to establish which input addresses of a transaction sent coins to a specific output address. Consequently, the ways of processing data are entirely different across networks. Thus, knowing the data types and network interactions is essential for implementing ML models on top of these networks.

## 3.2 Machine Learning Datasets and Learning Tasks

Chartalist is structured around blockchain types and currently serves UTXO and account-based blockchains. In the following, we describe a set of problems that can be addressed based on the datasets we provide and framed as blockchain graph ML tasks.

1. **Address Clustering**: This task aims to identify which addresses are co-owned by an entity [34]. To approach this task, protocol design choices and application-specific user behaviors may be exploited.

2. **Address and Transaction Type Classification**: This task aims to determine whether addresses and transactions belong to a specific class type. For example, certain addresses behave like cryptocurrency exchanges or certain transactions exhibit a malicious character [2, 35, 20, 23].

3. **Anomalous Transaction Pattern Detection**: This task aims to identify anomalous transactions so that, i.e., their distinct temporal patterns can be summarized. For example, ransomware payments typically involve similar, and the receiving addresses use certain transaction types [6, 21, 31, 38, 43].

4. **Multilayer Network Analysis**: This task aims to predict events and phenomena that emerge through the simultaneous use of multiple assets, which can be studied as multilayer networks [28].

5. **Coin Tracking Across Blockchains**: This task aims to uncover transaction flows between blockchains. For example, Bitcoins can be temporarily exchanged for privacy coins such as Monero or Zcash for obfuscation purposes.

6. **Price Analytics**: The goal of this task is to use observable blockchain activity, such as within the transaction network, to predict asset or coin prices [3, 1, 25, 18, 22, 11, 14, 10, 9] and the associated analysis of volatility [4, 42].

## 3.3 Tools and Frameworks for Blockchain Data Analytics

We curate our transaction network data by installing a blockchain node and connecting it to the P2P network. Afterward, we use a tool such as Bitcoin4J or Blockchain-ETL (`https://github.com/blockchain-etl`) to parse Bitcoin, Dash, and Ethereum. Although blockchain data can be accessed publicly, creating the transaction network is arduous. First, blockchain sizes have become prohibitively large to run data analytics in a single machine. Second, linking transactions to develop a blockchain network requires domain expertise, which many ML practitioners lack. Third, as BlockSci warns "cryptocurrencies continue to update their protocols, they may lose compatibility with BlockSci parsers". Chartalist aims to be a reliable supplier of blockchain graphs and meet a vital role in graph machine learning to combat this. Many blockchain analytics companies offer REST APIs to access blockchain data and networks. Examples are etherscan.io, blockcypher.com, and infura.io. However, APIs are costly or allow limited access to the API through a rate limit. For example, etherscan.io allows only five queries per second for the Ethereum network.

We provide datasets for the aforementioned tasks on Bitcoin, Dash, and Ethereum. **With the community's help, we aim to provide a labeled dataset for each task on Bitcoin, Ethereum (and its asset networks), Monero, Zcash, Dash, and Ripple** and continuously expand the number of blockchains covered. All datasets contain transaction time, metadata, provenance information, and curated labels. Chartalist datasets vary in size between 50 thousand and 40 million edges.

In addition to transaction graphs, we have curated address labels for the tasks. They originate from the domains of ransomware attacks, asset networks, and decentralized finance. In particular, we have used the label cloud of `etherscan.io` to curate Ethereum address labels (see [35]). We have curated our Bitcoin ransomware address labels from Montreal [30], Princeton [17] and Padua [12] studies. We have provided our code and data under a CC BY-NC (Attribution-NonCommercial) license.

**Limitations.** New Blockchain addresses are created daily. Hence new address labels may be missing from our labeled Ethereum dataset. On Bitcoin, many entities pay the ransom without disclosing the payment [15]. As such, our ransomware dataset cannot be complete. Our goal in sharing the dataset is to encourage detecting such undisclosed payments.

**Ethics and Privacy.** We must stress that all transaction data is theoretically already available to users with the resources (fast SSDs, large RAM, and disk space) to synchronize blockchain clients with the respective networks and manually extract information. An individual user's privacy depends on whether personally identifiable information (PII) exists about any of their blockchain addresses, which serve as account handles and are understood to be pseudonymous. Our data does not contain any PII. However, it is possible that if such PII were to be obtained from other sources, the datasets we are proposing here could lead to further privacy-related implications. This scenario cannot be prevented *because all raw data is already accessible to users controlling significant hardware resources*. Real-life identities may be discovered by using IP tracking information, *which we do not have nor share*. Furthermore, we curate all labels from public information or publicly available models (e.g., etherscan.io tags for Ethereum). The data does not contain individually identifiable information or personal attributes. Hence, there is no need for any anonymization of the data.

Most of our labels are published by Blockchain entities themselves. For example, exchanges publish their Ethereum addresses. Ransomed entities publish Bitcoin ransom addresses. Note that our Bitcoin label dataset consists of solely such disclosed addresses. We, for example, have not released a set of suspicious (potentially ransomware) addresses that we found in our previous work [2] for privacy reasons. Furthermore, we provide an online tool to appeal for the exclusion of an address from our dataset with a promise of responding to such requests promptly.

## 4 Tasks and Baseline Experiments

This section describes the first three tasks in more detail and provides baselines for them. We refer the reader to our website for more details on the other tasks.

### 4.1 Ransomware Address Classification on Bitcoin

Blockchain transactions can be created anonymously, and participation in the network does not require identity verification. Hackers can demand a ransomware payment by delivering a public blockchain address (i.e., a short string) to a sender using anonymity networks such as Tor [13].

**Why is the task important?** Malicious actors have noticed blockchain technology's ease of usage and worldwide transaction availability. The pseudo-anonymity of users in cryptocurrencies such as Bitcoin has attracted the interest of a diverse body of criminals, transnational terrorist groups, and illicit users. Cryptocurrency-related crime and criminal abuse of blockchain technologies are nowadays recognized as the fastest-growing type of cyber-crime. Ransomware has emerged as a critical threat to infrastructure in many countries. Using cryptocurrencies for ransomware payments appears to be substantially more prevalent than has been previously realized. As noted by Hernendex-Castro et al. [15], among the respondents to their survey, "the prevalence of the CryptoLocker ransomware seems much higher than expected". Detecting undisclosed payments and discovering ransoming actors have become critical tasks in blockchain data analytics.

Ransomware analysis has two sub-problems: identifying undisclosed payment addresses of a known ransomware family and detecting the emergence of a new ransomware family. We formulate and include the second problem as a particular case of the first one and formalize the problem as node label prediction in a directed weighted graph.

**Experimental Setup.** This task uses the Bitcoin transaction graph in Chartalist. Using a time interval of 24 hours, we extracted daily transactions on the network and formed the Bitcoin graph. We have filtered out the network edges that transfer less than $BTC$ 0.3 for computational efficiency since ransom amounts are rarely below this threshold. We have labeled 24,486 addresses from 26 ransomware families (see [2]).

Let $f_1, \ldots, f_n$ be labels of known ransomware families observed until time point $t$. We set $f_0$ as the label of addresses that are **not** known to belong to any ransomware family, and we assume them to be **white addresses**. Let $rs$ be a known ransomware family of interest. Let $\tilde{Y}_t \subseteq Y_t$ be

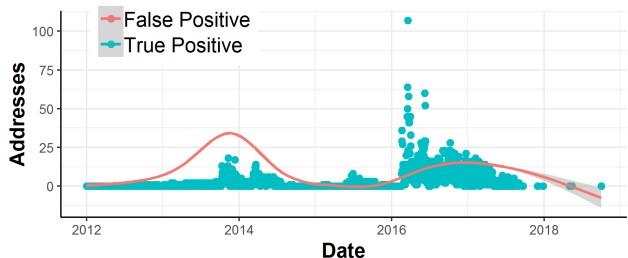

Figure 2: Ransomware detection with exact feature matches.

such that $\forall y_j \in \tilde{Y}_t, y_j \in \{f_0, f_{rs}\}$. Let $\{a_{l+1}, \ldots, a_{l+z}\}$ be a set of addresses whose set of labels $Y_{t'} = \{y_{l+1}, \ldots, y_{l+z}\}$ is unknown. Let $t' > t$, and $t < \min\{t_{a_{l+1}}, \ldots, t_{a_{l+z}}\}$. The problem is to predict all addresses $a_m \in \{a_{l+1}, \ldots, a_{l+z}\}$ such that $y_m = f_{rs}$, using the transaction graph and history $(X_t, Y_t)$.

**Baseline.** We approach the baseline detection task by extracting six features [2] from the daily Bitcoin transaction network for each address. We have designed the graph features to quantify ransomware operators' specific obfuscation patterns. Afterward, we employ tree bases methods, clustering, and naive similarity search on the feature matrix of all Bitcoin addresses.

Address clustering identifies which blockchain addresses are controlled by the same real-life entity/person. A popular method involves finding input and output address associations in a transaction. For example, on Bitcoin, two heuristics are widely used as baselines [24]. **Co-spending heuristic:** "If two addresses are inputs to the same transaction, the same user controls them". **Transition heuristic:** "If we observe one transaction with addresses A and B as inputs and another with addresses B and C as inputs, then we conclude that A, B, and C all belong to the same user". Using co-spending and transition heuristics with all history, we discover only 40 unique addresses from CryptoLocker, CryptoWall, CryptoTorLocker2015, CryptoTorLocker2015 families.

*Naive similarity search*: An interesting benchmark for detecting undisclosed payments from existing families is to compute the similarity of $X_{t'}$ addresses to the past addresses in $X_t$. If existing families exhibit repeating patterns over time, the similarity search can match new addresses to known ransom addresses. Figure 2 shows that this strategy may be effective. Using exact matches to known ransom patterns of the past, a six-day similarity search reveals more than 50 addresses each day while predicting a maximum of 73 false positives on 2013, day 336. These results offer supporting evidence of the utility of our features. However, this naive approach creates 21,371 FP addresses for the period we considered, making it unfeasible for operational use by security analysts.

**Results.** As Table 1 shows, baseline models yield better recall than precision. Similar to our (proposed) topological data analysis method (TDA), which performs the best, the DBSCAN clustering algorithm can ignore data points in its model building; two of the best non-TDA results are delivered by DBSCAN models. In the best TDA models for each ransomware family, **we predict 16.59 false positives for each true positive**. In turn, this number is 27.44 for the best non-TDA models.

Table 1: Detecting undisclosed ransomware payment addresses.

| RS | Method | TP | FP | FN | TN | Prec | Rec |
|---|---|---|---|---|---|---|---|
| Locky | TDA | 451 | 2350 | 50 | 8221 | 0.161 | 0.900 |
| | COSINE | 2395 | 41681 | 3990 | 146369 | 0.054 | 0.375 |
| Crypto | TDA | 217 | 3087 | 155 | 11200 | 0.066 | 0.583 |
| Wall | DBSCAN | 728 | 18960 | 794 | 16913 | 0.037 | 0.478 |
| Crypto | TDA | 439 | 9686 | 212 | 22129 | 0.043 | 0.674 |
| Locker | DBSCAN | 935 | 42771 | 295 | 11316 | 0.021 | 0.760 |
| Cerber | TDA | 187 | 5174 | 459 | 23027 | 0.035 | 0.289 |
| | XGBOOST | 1606 | 47307 | 7279 | 374169 | 0.033 | 0.181 |
| Crypt | TDA | 77 | 2460 | 271 | 11057 | 0.030 | 0.221 |
| XXX | COSINE | 589 | 20872 | 610 | 42952 | 0.027 | 0.491 |

## 4.2 Address Classification on Ethereum

The Ethereum project [40] was created in July 2015 to provide smart contract functionality on a blockchain. Smart contracts are Turing-complete software codes that are replicated across a blockchain network. Some smart contracts implement mechanisms to allow the trading of digital assets, known as tokens [36], on the blockchain. We refer to such a smart contract as an *asset contract* and use the term asset interchangeably. Like cryptocurrencies, an asset is transferred publicly between accounts (addresses). It may have an associated value in fiat currency which is arbitrated by asset demand and supply in the real world. These asset transfers create a directed, weighted multigraph between Ethereum addresses.

**Why is the task important?** Blockchains have enabled a new class of decentralized services, such as lending protocols, data oracles, stablecoins, and decentralized exchanges. The entities can employ regular accounts or smart contracts to facilitate information exchange or asset trades among blockchain addresses, lend coins, and provide a forum for financial decision-making. Although blockchain user addresses are pseudonymous, some entities publish or confirm their addresses to foster trust and security in their communities. Furthermore, analytics companies curate addresses of blockchain entities to analyze blockchain dynamics in real-time. However, manual curation is time-consuming, costly, and error-prone in the decentralized finance ecosystem that trades billions of US dollars worth of assets.

By modeling transaction behavior, we automate address identification and address type (e.g., proxy address of an exchange). Here we have formulated this task as identifying the most central nodes in Ethereum token networks (see [35]). Token networks offer more; we have also created a multilayer network out of token networks to detect global temporal anomalies (e.g., when blockchain technology is banned in a country). However, due to space limitations, the anomaly task is not explained here (see [28]).

**Experimental Setup.** This task uses the token networks dataset in Chartalist. Our address labels are taken from Etherscan,[1] a prominent Ethereum block explorer. Covering 149 centralized and decentralized exchanges, we have manually collected 296 account addresses that were listed publically in May 2020. Labeled data is scarce; not all the top token networks have multiple exchange nodes taking part in them. For the following evaluation, we only consider those token networks that contain at least ten such labeled exchange nodes, which reduces the number of networks considered to 28. We first evaluate the top 100 networks by the number of transfers.

We evaluate the task of finding top $n$ labeled nodes in token networks, using precision and recall measures (shown as $P@n$, $R@n$ respectively), where $AP@n$ and $AR@n$ denote the averages across multiple networks. Nodes are ranked by their core membership in descending order.

**Baseline.** We employ $k$-core and weighted $k$-core decomposition as the baseline and add our recent AlphaCore [35] decomposition results for comparison.

Both $k$-core and weighted $k$-core are defined for undirected networks. As the token networks are directed multigraphs, we use a node's neighborhood size instead of their degree. For the other networks, degree equals neighborhood size. Furthermore, we compare against modified variations of $k$-core (rows 8 and 9 of Table 2) and weighted $k$-core (rows $11-13$), using different input features.

Traditional core decomposition algorithms propose little in alleviating data challenges. For example, weighted $k$-core suggests using $\alpha = \beta = 0.5$ for combining properties, but improving over this arbitrary choice is left as future work. Most networks' edge weights are skewed, and long tails complicate scaling and normalization issues. For example, max edge weights in token networks can reach $10^{13}$, whereas node degrees are typically less than 3. With such a difference in scales, node property combination becomes a challenge in weighted $k$-core.

**Results.** The first insight is that data depth-based AlphaCore [35] performs best compared to the other two core decomposition algorithms. The results indicate that addresses of centralized and decentralized exchanges can be discovered by using in and out-degrees of addresses. This is a promising result to automate address type discovery in Ethereum asset networks.

---

[1] https://etherscan.io/labelcloud

Table 2: Performance comparison with a ranking task using 28 token networks. Performance is indicated using (average) precision and recall at $k$. We compare core decomposition methods with a variety of input features. Feature $N_{in}$ refers to the number of neighbors with incoming edges, and $S_{in}$ to the sum of weights (strength) of these edges.

|   | Algorithm | Input features | AP@10 | AP@20 | AP@50 | AR@10 | AR@20 | AR@50 |
|---|-----------|----------------|-------|-------|-------|-------|-------|-------|
| 1 | **AlphaCore** | $N_{in}, N_{out}$ | **0.54** | **0.45** | **0.30** | **0.10** | **0.16** | **0.25** |
| 2 | **AlphaCore** | $N_{out}$ | 0.50 | 0.44 | 0.27 | **0.10** | **0.16** | 0.23 |
| 3 | **AlphaCore** | $N_{out}, S_{in}, S_{out}$ | 0.42 | 0.33 | 0.21 | 0.07 | 0.11 | 0.18 |
| 4 | **AlphaCore** | $N_{in}, N_{out}, S_{in}, S_{out}$ | 0.41 | 0.33 | 0.22 | 0.07 | 0.11 | 0.19 |
| 5 | **AlphaCore** | $N_{in}$ | 0.50 | 0.41 | 0.24 | 0.10 | 0.15 | 0.21 |
| 6 | **AlphaCore** | $N_{in}, S_{in}$ | 0.39 | 0.30 | 0.19 | 0.06 | 0.10 | 0.17 |
| 7 | $k$-core | $N$ | 0.36 | 0.26 | 0.14 | 0.06 | 0.08 | 0.11 |
| 8 | $k$-core | $N_{out}$ | 0.22 | 0.16 | 0.11 | 0.03 | 0.05 | 0.10 |
| 9 | $k$-core | $N_{in}$ | 0.13 | 0.09 | 0.06 | 0.03 | 0.04 | 0.06 |
| 10 | w. $k$-core | $N, S$ | 0.37 | 0.30 | 0.19 | 0.06 | 0.10 | 0.16 |
| 11 | w. $k$-core | $N_{in}, S_{in}$ | 0.33 | 0.23 | 0.15 | 0.06 | 0.07 | 0.13 |
| 12 | w. $k$-core | $N_{out}, S_{out}$ | 0.28 | 0.29 | 0.21 | 0.05 | 0.10 | 0.18 |
| 13 | w. $k$-core | $N_{in}, N_{out}$ | 0.04 | 0.03 | 0.02 | 0.01 | 0.01 | 0.02 |

## 4.3 Edge Classification for Wash Trade Detection on Decentralized Exchanges

Decentralized Finance applications have emerged as one of the critical use cases of smart contract platforms like Ethereum. The complexity of the applications is increasing, and transactions are not only used as a means of value transfer but also to execute smart contract functionality.

**Why is the task important?** While the hashed representation of executed smart contract function signatures is observable, not all signatures can be translated to a human-readable form because there is a lack of publicly available smart contract source codes. Furthermore, single edges, as well as a collection of edges, may have a higher level of meaning that can range from simple actions such as permission changes to a set of transactions designed to perform arbitrage trades or malicious transactions designed to feign activity where there is none: for example by artificially inflating trading volumes by what is commonly known as wash trading. The latter represents an illegal activity in traditional markets where existing data is difficult to obtain. Stock exchanges usually do not provide information on which accounts have traded with each other. These examples illustrate the need for transaction or edge classification, especially in the Decentralized Finance setting. In the following, we focus on the wash trade detection example.

**Experimental Setup.** This task uses the anomalous transaction pattern detection dataset in Chartalist. Decentralized Exchanges (DEX) have grown very popular in the past couple of years and are usually implemented as smart contracts. There are several variants, but those most closely mimicking the centralized exchange model are Limit Order Book (LOB)-based exchanges. Users can place limit orders, directly accept existing orders, and trade one crypto-asset for another. However, as most DEXs do not implement Know Your Customer (KYC) procedures, users can easily wash trade crypto-assets. The Commodity Futures Trading Commission (CFTC) defines it as "entering into or purporting to enter into transactions to give the appearance that purchases and sales have been made, without incurring market risk or changing the trader's market position" [8]. This can be achieved on a blockchain by controlling multiple accounts and trading back and forth between them. The result is fake trading volume, which may draw the attention of unsuspecting investors.

Ground truth on wash trading activity on DEX does not exist. However, we have conservatively found trading patterns conforming to the wash trade definition. To find them, we have extracted all trades from blockchain transaction data of the first popular LOB DEX on Ethereum, namely IDEX and EtherDelta, up until May 4th, 2020. Each trade exchanges one crypto-asset for another, with information on participating accounts, amounts, and timestamps.

We have then constructed a directed, weighted trade graph, where the target node is always the one buying the native asset Ether in exchange for another crypto-asset. We then identify sets of edges

Table 3: Wash Trades Summary for IDEX and EtherDelta.

| Metric | IDEX | EtherDelta |
|---|---|---|
| # Wash Trades | 213,029 | 69,711 |
| Total Wash Volume USD | 83,531,254 | 75,846,518 |
| # Wash Trader Accounts | 659 | 5,533 |

that conform to the legal definition of wash trading, which is described in detail in our work [37] and outlined in the following paragraph.

**Baseline.** The wash trade detection task is approached with a two-step process. First, a candidate set of potential wash trades is determined by repeatedly identifying trade cycles in the form of strongly connected components, which are identified in multiple time windows. In a second step, the candidate set is evaluated on whether each participating account's position (asset balances) has not, or almost not, changed. Performing the second step can be used as an evaluation criterion when a different candidate set generation mechanism is employed. For example, one could devise a random walk strategy [29] or use graph convolutional networks to identify wash trades. A ML-based approach might learn the same type of patterns that we have manually devised in the original work. To compare with the baseline, the results should additionally be checked for whether they conform to the legal definition of no position change. The primary score is then the number of identified wash trades.

**Results.** Table 3 displays baseline results obtained through the repeated identification of strongly connected components in multiple time windows. We can determine the number of wash trades that meet the legal definition. We can also resolve the total wash trade volume and the number of involved trading accounts. These lower bound numbers can be improved with more advanced methods.

## Broader Impact

E-Crime involving crypto has recently witnessed an unprecedented surge in volume and variety, ranging from speculative behavior to money laundering to human trafficking. By preying on people's fear, one of the most recent frauds includes luring users to download ransomware, mimicking the COVID-19 dashboard tracker, and then asking for payment in cryptocurrencies. During the first five months of the COVID-19 pandemic in 2020, criminal activities on blockchain resulted in $1.36 billion. Deterring and combating such malicious activities in cryptocurrency blockchains' transparent environment requires developing novel data-driven approaches to threat detection on blockchain networks. Our dataset is one of the first steps toward facilitating the development of systematic and versatile ML tools for blockchain data analytics and detecting malicious activity on dynamic blockchain networks. However, it can also be viewed as a double-edged sword. The ML tools we discuss can assist law enforcement agencies in better tracking suspicious patterns in cryptocurrency trades. They can enhance their strategies to identify and combat financial crime on a broader security front– thereby, addressing issues of critical societal importance. However, the discussed methodology can be equally used by cyber-criminals to develop more efficient obfuscation schemes. For example, armed with the knowledge of which trading patterns and associated topological footprints are trackable in practice over time, criminals can elaborate schemes allowing them to hide their identity with much higher certainty.

## 5   Conclusion

In the past ten years, blockchains have gathered a wide variety of data from all walks of life. However, ML and data science methods for analyzing blockchain data tend to be noticeably delayed than the blockchain technology itself. Some domains of blockchain data analytics, such as ransomware payment detection in cryptocurrencies, have emerged as a fundamental societal problem, demanding the development of novel machinery of ML tools. Chartalist makes blockchain data more accessible to the broader ML community and, as such, stimulates the development of innovative ML tools for blockchain data analytics. Chartalist is the first comprehensive platform for blockchain graph datasets with ground truth suitable for a broad range of ML research tasks, from node classification to time-series forecasting.

## Acknowledgments

This material is based upon work sponsored by the Canadian NSERC Discovery Grant RGPIN-2020-05665: **Data Science on Blockchains**, the National Science Foundation of USA under award number ECCS 2039701 **Blockchain Graphs as Testbeds of Power Grid Resilience and Functionality Metrics**.

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
