# OpenReview forum: "Chartalist: Labeled Graph Datasets for UTXO and Account-based Blockchains"
_NeurIPS.cc/2022/Track/Datasets_and_Benchmarks — NeurIPS 2022 Datasets and Benchmarks _

### Official Review · Reviewer_puuZ · 2022-07-25
**A great contribution to the Blockchain ML space**

**Rating:** 8
**Confidence:** 3
**Correctness:** Yes.
**Clarity:** Yes. The paper is well written and ea…

**Strengths:**

The authors are commended for making blockchain data more accessible to those who wish to perform graph ML analysis. This is achieved by providing cleaned and curated blockchain-related datasets, as well as extraction and graph creation tools to facilitate additional data collection. The motivation and methodology for each sample task is well explained.

**Weaknesses:**

The paper makes several mentions of Machine Learning benchmarks, including an entire section in the literature review, however the benchmark capability of Chartalist is never outlined or introduced directly in the paper.

**Additional Feedback:**

Excited to see where this project goes.

**Documentation:**

The paper and associated package (via Github) is well documented, however the website at chartalist.org feels slightly lacking in this regard.

**Ethics:**

Ethical concerns are well addressed.

**Relation To Prior Work:**

Yes.

**Summary And Contributions:**

This paper introduces a collection of blockchain graph datasets and a platform to access them. Several potential applications are explored and assessed. Additional tools are provided to facilitate further data collection.

---

> ### Author Response · Authors · 2022-08-23
> **Response to Reviewer puuZ**
>
> Thank you so much for the very constructive and motivating suggestions on the project improvement and also for appreciating the novelty and contributions of Chartalist.
>
> Thank you so much for the very constructive and motivating suggestions on the project improvement and also for appreciating the novelty and contributions of Chartalist. Since we have different sections and datasets we summarize our paper due to the space limitations. To make the capabilities clearer we have designed a figure in which we summarize all the capabilities based on the datasets. We updated the main page of our website with this figure to make the users understand our capabilities faster and easier. We hope that these changes have resulted in a more cohesive and clear outline of the proposed datasets and the associated capabilities.

---

### Official Review · Reviewer_xh75 · 2022-07-26
**Compilation of datasets from blockchains to spur machine learning research for various use cases**

**Rating:** 7
**Confidence:** 3

**Strengths:**

The work published in this paper can have a significant contribution in attracting a wide community of scientists to analyze blockchain data. Nature of pseudo-anonymity of users in cryptocurrencies and a rapid increase in transactional activities on blockchain is attracting cyber criminals to perform ransomware attacks, money laundering or price manipulation. ML has been widely adopted by centralized service providers to prevent cyber crime and so can be used in blockchain based decentralized service providers to make the platforms more secure. These datasets serve the purpose of spurring the required research by ML community on making the blockchain based applications more secure for users.

The datasets have been made accessible with the help of easy to use data loaders. The authors take accountability in hosting, managing, and updating the data repository at a quarterly basis at-least. The datasets do not contain PII and has no privacy related concern other than what is already available publicly.

**Weaknesses:**

One weakness that is already quoted by the authors in the paper is the potential negative use of the datasets by malicious actors to learn to hide their identities without getting detected. The second weakness of the paper is the lack of details on label generation process beyond the fact that they were sourced from etherscan.io and from past studies. The third weakness is lack of recommended benchmarking metrics for proposed ML tasks. For example, in Table 1, results on standard classification metrics are discussed, but a recommendation around the choice of score threshold to convert soft score to hard labels is missing.

**Additional Feedback:**

Suggestions for improvement:
1) Detailed explanation of label curation in different datasets, 2) Sample datasets in Appendix, 3) Suggested performance metrics for benchmarking.

**Clarity:**

Most of the paper is well written and the authors have provided enough context for the readers so they can understand different blockchain related terminologies wherever required.

In certain sections, the authors have scope of improvement. For example, Table 3 lacks explanations of results. The number of wash trades that were identified from the first step vs. the second step is not discussed. The analysis depicting results in Table 3 uses proxy labels but how the labels were generated is not clearly explained. As another example, the experimental setup in Section 4.2 is also not well explained. If the number of networks under consideration were reduced to 28, then what defined top 100 networks.

**Correctness:**

To the best of my understanding, the datasets are constructed in a sound way. The transaction network data is curated by connecting blockchain node to P2P network and using ETL tool to parse Bitcoin, Ethereum and Dashcoin.

**Documentation:**

The documentation of data, meta data, data loader API is very well written.

**Ethics:**

One possible ethical concern includes an easier access of transaction data to all users, including ones with malicious intent, who can use it to create non traceable attack vectors. This concern is already acknowledged by the authors in the paper. The datasets do not contain PII. So there is no privacy concern.

**Relation To Prior Work:**

It is clearly mentioned in the paper that this is the first attempt to systematically organize blockchain data for the broader ML community. Other methods to analyze blockchain data that include running a Bitcoin client or paying for commercial API are also discussed. Previous contributions are discussed in relations to data repositories, ML benchmark and blockchain analysis tools.

**Summary And Contributions:**

The authors launch Chartalist, a platform to access a variety of datasets from blockchain that can be used for ML research tasks. The datasets are currently compiled from Bitcoin, Ethereum and Dashcoin blockchains. The datasets are well organized and documented in the official website, and easy to use data loaders to pull datasets using Pandas in Python are provided. These datasets can be used for various ML tasks including but not limited to Address Clustering, Address and Transaction Type Classification, Anomalous Transaction Detection etc. Baselines and model comparisons on three such tasks are discussed in the paper. The baselines are helpful to allow researchers get up to the speed with simple modeling approach and allow them to try new ideas to improve the performance. The datasets are partially labeled. The tool and the datasets are open source and authors promise to maintain them. Most of the datasets can be represented in the form of graph network and graph ML techniques can be employed.

The main contributions and novelty in this work is making a large scale, hard to compile blockchain dataset accessible to the machine learning research community. The datasets can be easily loaded for any analysis or experimentation for ML task using Python.

---

> ### Author Response · Authors · 2022-08-23
> **Response to Reviewer xh75**
>
> We are very grateful to the reviewer for the detailed and constructive feedback and especially for the very thought-provoking questions on data privacy! We are also very thankful for the reviewer's appreciation of the novelty and potential of Chartalist.
>
> 1. One weakness that is already quoted by the authors in the paper is the potential negative use of the datasets by malicious actors to learn to hide their identities without getting detected.
>
> Thanks very much for raising this important question. Yes, we believe that this issue is applicable for many adversarial machine learning relate tasks and datasets. Still making these public datasets available to wider ML community could result in the creation of more accurate models for detecting malicious activity on blockchains. As it is, the accuracy of the existing ML models are not high enough, and more research is urgently needed.
>
> 2. The second weakness of the paper is the lack of details on label generation process beyond the fact that they were sourced from etherscan.io and from past studies.
>
> We agree with this concern, however, in fact, the label procurement process was no more complex than manually collecting public, non-login restricted data from Etherscan. Following the reviewer's question, we have now specified in Section 4.2 that the collection was performed in May 2020) and aggregating known labels from existing papers.
>
> 3.  The third weakness is lack of recommended benchmarking metrics for proposed ML tasks. For example, in Table 1, results on standard classification metrics are discussed, but a recommendation around the choice of score threshold to convert soft score to hard labels is missing.
>
> Thank you for your comment. In our experiments for Dbscan and Cosine, we use clustering association for class labels. Similarly, TDA creates a topological network where vertices are soft clusters of addresses. As a result, for Dbscan, Cosine and Tda, we do not use thresholds. Only in Xgboost, tree leaves use a default 0.5 probability threshold which we kept. Because of space limitations we couldn't put more details in this section. For performance details of parameters, please see tables 1 and 3 of the Bitcoinheist article. (reference [1])
> More detail about each datasets and benchmarks are available on our website.

---

> > ### Comment · Reviewer_xh75 · 2022-08-25
> > **Clarification on "recommendation around the choice of score threshold"**
> >
> > >Only in Xgboost, tree leaves use a default 0.5 probability threshold which we kept
> >
> > To clarify, by recommendation about the choice of threshold tuning, I meant a discussion on according to the authors, what should be used as a pivot point to tune the thresholds, or what kind of trade-off between metrics (e.g. precision, recall, false positive rate) should one expect. Using a default threshold of 0.5 from non calibrated models does not have a practical value.

---

> > > ### Author Response · Authors · 2022-08-25
> > > **Response to Reviewer xh75**
> > >
> > > Thank you for the clarification. As the blockchain data is always quite imbalanced (i.e., there are few known address or transaction labels, but millions of ordinary addresses and transactions), the precision/recall trade-off is always a concern. We use back-testing as a result verification strategy in ML models. In back-testing we consider two use cases of our ML models. If an analyst is mining  blockchain data for e-crime, it may try to manually investigate fewer addresses and transactions, because the network size is too big to analyze everything manually. In this case, precision is the primary concern. If a bank is analyzing the data, its analysts may want to flag every suspicious data piece to avoid future law suits for abetting money laundering and wash trading. Recall is the primary concern here.
> > >
> > > In address and transaction analysis (i.e., BitcoinHeist), we used two TDA Mapper parameters to control the precision/recall trade-off: inclusion and size. The inclusion parameter limited what can be learned when very few ransomware addresses are contained in the cluster. The size threshold prevented learning when cluster includes too many (negative or positive) addresses.
> > >
> > > In our experience, precision is more useful than recall in blockchain data, because ML results need to be further manually analyzed, and the analysis cannot be completed when the blockchain data is too big and finds too many (false or true) positives.

---

> > > > ### Comment · Reviewer_xh75 · 2022-08-26
> > > > **Response to the authors**
> > > >
> > > > The discussion in your response makes sense to me. Please add this to the paper, if possible.

---

> > > > > ### Author Response · Authors · 2022-08-26
> > > > > **Thank you!**
> > > > >
> > > > > Thank you, we certainly will.

---

### Official Review · Reviewer_tG6T · 2022-07-26
**Chartalist provides a platform that hosts a large set of datasets from different blockchain-based cryptocurrency networks that can facilitate different ML tasks.**

**Rating:** 6
**Confidence:** 5
**Clarity:** The paper should be improved in at le…

**Strengths:**

The strength of Chartalist can be summarized as follows:
-	It contains useful datasets from three different cryptocurrency networks that are based on different architectural model (i.e. UTXO vs. account-based).
-	Based on each cryptocurrency network, it suggests several tasks that can be studied on each dataset, providing insights and direction to further research.
-	While there is a considerable lack of labeled data for analyzing blockchain data, Chartalist provides several datasets with labels including different kinds of heist activities.
-	Chartalist is open source, easy to use, and is supposed to be updated regularly.


**Weaknesses:**

The paper can be improved on the following aspects:
-	The organization of the paper can be substantially improved. The paper has a major issue with referencing; there is no reference for many of the statements and assumptions. Although many of these statements make sense to an expert in the field, there should be appropriate referencing for consolidating the ideas proposed.
-	The discussion in Section 3.1 does not follow naturally with the paper. Although it is important to discuss the blockchain data type, I think the discussion in this section is very long and does not connect well with the other sections. Also, it does not clearly state why UTXO and account-based architecture are different and why we need to consider the underlying differences when studying samples of these networks.
-	In Section 3.2, several different tasks have been introduced. However, the explanations for some of these tasks are unclear. It could help to improve the definitions of these tasks and provides some actual examples to better motivate the importance of these tasks.
-	It is mentioned that one important intuition behind Chartalist is to provide datasets for conducting ML/graph-based methods for several important tasks on cryptocurrency networks. However, in the experiments section, most of the baseline methods are basic and simple ML methods, and graph-based methods (such as methods based on GNN) are not considered.
-	Most of the methods are from the previous works of the authors and newer DL methods are not discussed in the evaluation of the datasets and the proposed tasks.
-	The results of the experiments are not discussed comprehensively. For example, it could be nice to discuss why the recall and precision reported in Table 1 are so low for several datasets and methods. Is there any relation between the datasets, method, and performance? Also, the reported results in Table 1 are so low; the paper should discuss whether achieving these low results is useful in any real-world scenario, or whether these kinds of experiments are actually useful in a real scenario?
-	The datasets contain temporal information but no tasks is defined to study the temporal aspect of the data.


**Additional Feedback:**

In total, I think Chartalist provides an important source of datasets for facilitating the analysis and investigation of blockchain networks. However, since many future works could consider this paper as an important related work, the paper should be improved in several aspects.

**Correctness:**

Although there are some assumptions about how the data is crawled from the blockchain (e.g., limiting the time window to be able to tackle the huge data size), the assumptions seem reasonable considering other related research.

**Documentation:**

The datasets and how they have been generated are explained and on the Chartalist website, different datasets can be found and easily accessed. However, the detailed explanation of how to crawl the data is not fully complete and can be improved for better clarity.

**Ethics:**

Since the blockchain is publicly available, there should not be serious issues. Also, on Chartalist website there is the possibility to exclude the address of a specific account if the user believe that it is incorrectly labeled.

**Relation To Prior Work:**

The related work should be improved, there are some other related platforms for the blockchain data analysis that have not been considered. One of the important examples is http://xblock.pro/#/ which also has several important datasets that are currently being used by related research. Also, the distinction of Chartalist with other dataset repository could be better explained.

**Summary And Contributions:**

This paper proposes Chartalist which is a platform that hosts a diverse set of datasets from three different cryptocurrency networks (Bitcoin, Ethereum, Dashcoin) that are useful for investigation and conducting research on blockchain networks. The datasets are ready to be used in different ML tasks and facilitate analysis of the transactions on cryptocurrency networks. The datasets contain transaction networks of three different cryptocurrencies and also encompass several different important tasks specific to each network.

---

> ### Author Response · Authors · 2022-08-23
> **Response to Reviewer tG6T 1/2**
>
> We would like to thank the reviewer for the detailed and constructive feedback, particularly on other DL/ML tools and the paper organization, and also for recognizing the novelty and potential of Chartalist.
>
> 1. The organization of the paper can be substantially improved. The paper has a major issue with referencing; there is no reference for many of the statements and assumptions. Although many of these statements make sense to an expert in the field, there should be appropriate referencing for consolidating the ideas proposed.
>
> Thanks you for your comment, We have added missing references to our paper for the final version. The revised version has been submitted.
>
> 2. The discussion in Section 3.1 does not follow naturally with the paper. Although it is important to discuss the blockchain data type, I think the discussion in this section is very long and does not connect well with the other sections.
>
> Thank you for raising this issue, Section 3.1 has been rewritten for better coherence.
>
> 3. Also, it does not clearly state why UTXO and account-based architecture are different and why we need to consider the underlying differences when studying samples of these networks.
>
> We have improved the description of Section 3.1 to reflect this concern and have entirely recreated Figure 1 to illustrate  the difference between UTXO and account-based blockchains.
>
> 4. In Section 3.2, several different tasks have been introduced. However, the explanations for some of these tasks are unclear. It could help to improve the definitions of these tasks and provides some actual examples to better motivate the importance of these tasks.
>
> Thank you for your comment, In section 3.2 we aimed to introduce all six different task which are available on our repository. We tried to explain the tasks briefly with some simple and straightforward examples. Since each task should be preformed on a specific datasets, we categorized all available tasks for each dataset on our website. Due to space limitation we could not add complete details for each task in the paper. Instead, we put more detailed discussion as well as data samples on our website for more information about each task. Furthermore, we added a flowchart to our website's main page, in order to introduce our benchmark capabilities based on each dataset in a visual manner.
>
> 5. It is mentioned that one important intuition behind Chartalist is to provide datasets for conducting ML/graph-based methods for several important tasks on cryptocurrency networks. However, in the experiments section, most of the baseline methods are basic and simple ML methods, and graph-based methods (such as methods based on GNN) are not considered.
>
> Thank you for raising this question as indeed blockchain data offer a
> wealth of opportunities for GNN benchmarking. We have performed a broad range
> of experiments with GNN baselines on Ethereum tokens [Chen et al., 2021] and
> [Chen et al., 2022]. (For the sake of space, these results are not included into the
> manuscript but are presented on the website.) However, in the case of Bitcoin, i)
> the dataset does not contain enough labeled data (we have less than 30K labeled
> addresses), and ii) the overall network is too large (600K vertices daily, for more
> than 10 years) to run many GNN methods. This is why we have opted for simpler
> ML models there.
>
> 6. Most of the methods are from the previous works of the authors and newer DL methods are not discussed in the evaluation of the datasets and the proposed tasks
>
> We agree with this concern but we would like to reemphasize that
> the primary goal of this paper is to introduce the ML audience to the current key problems in blockchain analytics which remain largely uncharted territory for ML scientists and to provide an easy one stop shop access to the blockchain data.
> For unlabelled Ethereum data, the rigorous experiments on GNNs and other DL have already appeared in ICML, ICLR. In turn, for labelled data, the experimental results have appeared in IJCAI, WebConf and KDD. Due to the space limitations, we cannot show all these results in the current manuscript which also has a somewhat different goal.
> In each dataset's article we compare our results to various models, such as the three heuristics in Bitcoin or known core decomposition tasks in Ethereum. That is, we already have extensive comparisons for each datasets. We hope that Chartalist datasets will be used extensively and new SOTA results will be achieved. We plan to prepare a leaderboard with SOTA results in the future.

---

> > ### Author Response · Authors · 2022-08-23
> > **Response to Reviewer tG6T 2/2**
> >
> > 7. The results of the experiments are not discussed comprehensively. For example, it could be nice to discuss why the recall and precision reported in Table 1 are so low for several datasets and methods. Is there any relation between the datasets, method, and performance? Also, the reported results in Table 1 are so low; the paper should discuss whether achieving these low results is useful in any real-world scenario or whether these kinds of experiments are actually useful in a real scenario?
> >
> > n our paper [Akcora et al., 2020], we used one of the datasets to de-
> > tect malicious addresses. Although the reported results seem low, they significantly
> > improved the state of art at the time. One reason these results are low is that
> > we have missing positive labels for many addresses, and in our research, we as-
> > sumed that unlabeled addresses are automatically benign. This may not be true
> > and in [Akcora et al., 2020], we provided further strong evidence that some bit-
> > coin addresses that are identified by our methods as ransomware related are indeed
> > ransomware. Hence, the reported results could be seen as a lower bound.
> > At the same time, we would like to stress that the existing results show the need
> > for further research on the topic, and the Chartalist dataset could encourage more
> > research.
> >
> > 8. The datasets contain temporal information but no tasks is defined to study the temporal aspect of the data.
> >
> > The temporal aspect is implied and vital in Blockchains in almost all tasks. Price
> > prediction models have found past price to be the most informative attribute.
> > For instance, Ethereum graphs offer a unique capability of intrinsically dynamic
> > graphs (i.e., edges, nodes and edge weight evolve in time) that can be used for
> > time-aware learning, including such tasks as node feature and graph forecasting,
> > link prediction and anomaly detection (these applications are discussed in our
> > ICML [Chen et al., 2022], ECML-PKDD [Ofori-Boateng et al., 2021],and ICLR [Chen et al., 202
> > Even in time agnostic applications, such as network core decomposition, Blockchain
> > researchers divide the transaction network into 24-hour snapshots (as the entire
> > network is too big) and study them in isolation. In another example, malicious
> > actors start using a ransomware money laundering pattern in time, and the ML
> > models should learn the origin of the model and apply it in future cases. In this
> > sense, blockchains are the most important temporal data source, and many models,
> > such as time series analysis and anomaly detection can benefit from the availability
> > of Chartalist data. Thanks to this review, we have now added a section to the network parsing help pages (e.g., https://chartalist.org/BitcoinData.html) to discuss
> > the temporal nature of the data.
> >
> > 9. In total, I think Chartalist provides an important source of datasets for facilitating the analysis and investigation of blockchain networks. However, since many future works could consider this paper as an important related work, the paper should be improved in several aspects.
> >
> > Thank you so much for your comments, we will make improvements in the mentioned sections.

---

> > > ### Comment · Reviewer_tG6T · 2022-08-26
> > > **Please reflect the changes to the paper**
> > >
> > > Thanks for updating the submission.
> > > I believe that the submission has improve from several aspects.
> > >
> > > However, I still have some concerns regarding the experiments and methods.
> > > In several cases, the authors mention that they have utilized the methods and experiments that are the direct results of the previously published works. In fact, this is not favourable, since it negatively affects the main contribution of the paper. Although the submission target "datasets and benchmark" track, it is expected to propose novel datasets and benchmarking suit (with SOTA methods being used as baselines). Presenting a collection of what have already been proposed does not bear enough contribution.
> > >
> > > In addition, the main modifications have been applied to the website. Although it is very important to have a good maintenance plan and host a comprehensive website, it is also important to have consistency in the paper, and to include all important information and discussion. Considering the page limit, the authors could include the important results and experiments (e.g., the ones related to GNN methods) in the appendix as supplementary material. Moreover, it is more interesting to include the results of more complicated ML/DL method (at least partially) in the paper, not only the simple methods.

---

### Official Review · Reviewer_Hrz4 · 2022-07-27

**Rating:** 6
**Confidence:** 4
**Correctness:** The claims are correct since they are…

**Strengths:**

1. The datasets are collected from real-world applications, which include unique patterns.
2. The potential tasks are listed clearly, and the website is well organized.
3. The potential real-world impact is illustrated thoroughly.

**Weaknesses:**

1. Although these datasets are evaluated by some existing methods, the performance of popular supervised/unsupervised architectures is unknown, e.g., GCN and Node2Vec. Is it challenging to evaluate these methods with the proposed graphs? Or do these popular methods fail on the mentioned tasks?
2. ``Some datasets are published somewhere else``. For example, ``Address Classification on Ethereum`` is included in ``AlphaCore: Data Depth based Core Decomposition``; ``Wash-Trading Transaction Identification Dataset`` is published in ``Detecting and quantifying wash trading on decentralized cryptocurrency exchanges``. I think it is okay to include these datasets, but it requires more discussions/experiments other than the published content.
3. I wonder if graphs' labels are good enough for a fair evaluation of different methods since it is difficult to get the ground truth.
4. I hope more details of datasets could be included in the paper or the supplementary but not only available on the website.


**Additional Feedback:**

I think more details should be included in the paper/supplement but not only available on the website.
The primary concern is that some datasets are published somewhere else. I think this paper should concentrate on what is new but not a summary of existing work like a short review. I appreciate the effort that the authors collect different datasets together. However, I think the contribution is somehow weak to publish a new paper unless there is a significant improvement over the published papers.

**Clarity:**

More details of the dataset could be included in a table, e.g., how many graphs in total, the size, visualizations of subgraphs, some basic graph properties.

**Documentation:**

The documentation website is well organized for downloading.

**Ethics:**

There are no ethical concerns.

**Relation To Prior Work:**

I think this paper is a summary of some published papers.

**Summary And Contributions:**

This paper proposed a set of graph datasets collected from blockchains. The background, importance, and potential applications are discussed. Three tasks are also evaluated with existing methods. The datasets are distinct from the widely used graphs such as citation networks and social networks. However, I wonder if this paper doesn't propose new datasets but gathers some existing graph datasets together. Some datasets are also published in other conferences and available on GitHub. Considering the main contribution of this paper is introducing new datasets but not comparing different existing methods, I think the contribution of this paper is somehow weak.

---

> ### Author Response · Authors · 2022-08-23
> **Response to Reviewer Hrz4**
>
> We very much like the reviewer's detailed and constructive feedback, Especially, questions on contrasting Chartalist with the existing data have led us to substantially improve our presentation. We hope that these changes would further convince the reviewer of the unique potential of Chartalist as a one-stop shop for blockchain data analytics.
>
> 1. [...] I wonder if this paper doesn't propose new datasets but gathers some existing graph datasets together.)
> Our approach is intentional. Please note that Chartalist follows the common practice in gathering ML datasets from some successful examples. For example, Stanford's very popular network repository (https://snap.stanford.edu/data/), the TU repository on graphs (https://chrsmrrs.github.io/datasets/) and many other repos provide datasets curated by researchers from diverse projects. In blockchains, it is very time and resource-consuming to extract the data from client applications and provide it in simple data formats. In addition to the basic transaction data in graph form, our data also consists of labels. Collecting these labels presents another challenge and often leads to new research using new labels, thus preventing comparability. This paper, therefore, aims to provide an entry point for researchers in the ML community. On the one hand, researchers can easily access blockchain datasets with Chartalist as a one-stop shop for blockchain data, and on the other hand, they can use a common repository to compare their results with baselines, ensuring that exactly the same datasets have been used.
>
> 2. [...] Is it challenging to evaluate these methods with the proposed graphs? Or do these popular methods fail on the mentioned tasks?
> as indeed blockchain data offer a wealth of opportunities for GNN benchmarking. We have performed a broad range of experiments with GNN baselines on Ethereum tokens [Chen et al., 2021] and [Chen et al., 2022]. (For the sake of space, these results are not included in the manuscript but are presented on the website.) However, in the case of ransomware
> detection on Bitcoin, i) the dataset does not contain enough labeled data (we have
> less than 30K labeled addresses), and ii) the overall network is too large (600K
> vertices daily, for more than 10 years) to run many GNN methods. This is why we
> have opted for simpler ML models there.
>
> 3.  [...] I think it is okay to include these datasets, but it requires more discussions/experiments other than the published content.
> Unfortunately, due to page number limitations, we could not include all the details about different tasks. Furthermore, we do not want to re-run all the experiments already reported in the published papers. Our goal here is different; that is, we aim to provide sufficient details to show the reader what is possible to run different ML tasks with the provided datasets. For each included section, we added the reference to the existing papers.
>
> 4.  I wonder if graphs' labels are good enough for a fair evaluation of different methods since it is difficult to get the ground truth.
> Some of the labels are generated by the community and/or security researchers. Similar to other existing datasets, it is possible that some of those labels may be erroneous. Still, the dataset labels available represent state of the art, and in many cases those labels are generated using strong ground truth information (e.g., a cybersecurity researcher is sending small funds to a Bitcoin address after executing and observing a real ransomware behavior). At the same time, we agree that these datasets by themselves should not be considered the "ultimate" benchmark in comparing ML algorithms, and the performance of ML algorithms should be evaluated across multiple benchmarks. By sharing these blockchain datasets, we increase the availability of existing real-world benchmark datasets.
>
> 5. I hope more details of datasets could be included in the paper or the supplementary but not only available on the website.
> Our data is either tabular or graph data. We added visualized token network graphs on our GitHub library and also added key figures and samples to the appendix of the paper.
>
> our aim was to provide a starting point for researchers in the ML community. It is true that most of these datasets
> are already available elsewhere, but the information is spread out and not presented
> in the context of benchmark tasks and existing baselines. With Chartalist, on the
> one hand, researchers who want to get started with these blockchain-specific tasks
> can now easily access blockchain datasets, and on the other hand, they can use a
> common repository to compare their results with baselines, ensuring that exactly
> the same datasets have been used. The latter is a common pitfall, as individual data
> extraction usually leads to different choices of selected time ranges and obtainable
> labels. In addition, we have also included a novel dataset (Dashcoin) and plan to
> further update them with the help of the community.

---

> > ### Comment · Reviewer_Hrz4 · 2022-08-24
> > **Thanks for the response!**
> >
> > Thanks for the response! However, the answer doesn't fully address all my concerns. So I hold my original score.
> > 1. I agree that the paper could include existing datasets, and I understand that collecting graphs from blockchain is complicated. But I think some new analysis is needed to publish a full paper in this case. I think the paper should include content such as the property of these graphs, the challenges for conventional methods, and which property should be considered for successful estimation/clustering/detection.
> > 2. Space limitation is not an issue. The author could include all other content in the supplementary.
> > 3. Since the content and the analysis are included in other papers, what are this paper's novelty and contribution? Did authors clean or standardize the original dataset? I think the authors should clarify what is new in this paper instead of reporting the existing results from other papers. I think the contribution is somehow weak if most datasets are available elsewhere and the authors only collect them in the same place.
> > 4. I think it is not good enough for a full proceeding but would be a clear acceptance for a workshop.

---

> > > ### Author Response · Authors · 2022-08-25
> > > **Response to Reviewer Hrz4**
> > >
> > > We thank the reviewer for the comments. We have updated the website and the manuscript accordingly. Please find our changes and responses below.
> > >
> > > 1- I agree that the paper could include existing datasets, and I understand that collecting graphs from blockchain is complicated. But I think some new analysis is needed to publish a full paper in this case. I think the paper should include content such as the property of these graphs, the challenges for conventional methods, and which property should be considered for successful estimation/clustering/detection.
> > >
> > > Challenges: For each task, we have now added a section on the website to describe the challenges of working on the task for the related dataset. Properties: In the appendix, we have given a list of node properties for the Ethereum address classification task. For Bitcoin, all six features had been found useful for address classification; we have added their description and design rationale to the appendix. For graph properties, we have now included a figure on the number of unique addresses and transactions in time for Bitcoin.
> > >
> > > 2- Space limitation is not an issue. The author could include all other content in the supplementary.
> > >
> > > The leaderboard results have been included on the website and in the appendix. In fact, we have extended the appendix considerably with all the reviews.
> > >
> > > 3- Since the content and the analysis are included in other papers, what are this paper's novelty and contribution? Did the authors clean or standardize the original dataset? I think the authors should clarify what is new in this paper instead of reporting the existing results from other papers. I think the contribution is somehow weak if most datasets are available elsewhere and the authors only collect them in the same place.
> > >
> > > The main novelty and contribution (and the difficulty) of Chartalist are the transaction graphs which cannot be parsed and output-linked on off-the-shelf computers. Please note that we share the entire transaction network of Bitcoin and Token networks of Ethereum and promise to keep the data up to date in time. Although labeled datasets have appeared in earlier publications, the graph data that we share cannot be found on the Internet - no service gives the whole data publicly and without a cost. Chartalist will allow researchers from distant corners of the world to embark on blockchain data analytics.

---

> > > > ### Comment · Reviewer_Hrz4 · 2022-08-25
> > > > **Could you list which dataset is published and which dataset is unavailable?**
> > > >
> > > > It is essential to point out which dataset is published and which is not and what analysis is new. There are three tasks listed in the paper. I can find two of them from Github, and all the analyses are published elsewhere, which is why I think the contribution is weak in this paper.
> > > >
> > > > For example, https://github.com/friedhelmvictor/alphacore includes the dataset "Address Classification on Ethereum". This paper includes the same result as reported in "Alphacore: Data Depth Based Core Decomposition". So I believe the dataset and the analysis are the same.
> > > >
> > > > "Detecting and Quantifying Wash Trading on Decentralized Cryptocurrency Exchanges" has the same results as this paper. And the dataset is available here: https://github.com/friedhelmvictor/lob-dex-wash-trading-paper.
> > > >
> > > > I didn't find the dataset for "Ransomware Address Classification on Bitcoin", but the analysis and the results are the same as reported in BitcoinHeist: Topological Data Analysis for Ransomware Detection on the Bitcoin Blockchain (https://arxiv.org/pdf/1906.07852.pdf).

---

> > > > > ### Author Response · Authors · 2022-08-25
> > > > > **Response to Reviewer Hrz4**
> > > > >
> > > > > Thanks for your comment, In addition to our previously shared data, we share the following datasets that have never been published before because extracting data and creating the network requires considerable resources; Bitcoin data is over 350GB on disk, and linking outputs to addresses requires at least 60GB memory. Ethereum storage is similarly large, and transactions must be executed in the Ethereum Virtual Machine to extract token, stablecoin and transaction networks. These challenges are the primary reason why blockchain data have not yet propagated widely into the ML community.
> > > > >
> > > > > Bitcoin: Entire transaction network (https://chartalist.org/BitcoinData.html) from 2009 to 2022. Bitcoin has more than 40M vertices and 1.6B transactions. Please see Appendix Fig 1 for vertex and transaction counts in time.
> > > > >
> > > > > Ethereum:
> > > > > 1) Stablecoin transaction networks (https://chartalist.org/eth/StablecoinAnalysis.html) of USDT, USDC, DAI, UST, PAX and WLUNA. These are the most popular stablecoins on Ethereum by volume, and contain 5M edges (transactions).
> > > > > 2) Token transaction networks (https://chartalist.org/eth/TaskMultilayer.html) of 1701 most popular ERC20 tokens by volume. The networks contain 9.9M unique vertices (addresses) and 4M edges (transactions).

---

> > > > > > ### Comment · Reviewer_Hrz4 · 2022-08-25
> > > > > > **Thanks for the clarification!**
> > > > > >
> > > > > > The authors addressed my major concerns, so I increased my score. But I think the writing should be improved.

---

> > > > > > > ### Author Response · Authors · 2022-08-26
> > > > > > > **Thanks very much!**
> > > > > > >
> > > > > > > Thank you very much for the interactive and stimulating discussion which has resulted in many important improvements of the paper  and, of course, for increasing the score! We will certainly polish the writing for the camera ready version.

---

### Official Review · Reviewer_4zbJ · 2022-07-27
**Chartalist**

**Rating:** 6
**Confidence:** 3

**Strengths:**

This work reduces the burden of preprocessing the raw blockchain data which needs a lot of time and resources. And it provides the labeled datasets of different tasks on blockchain graph research and a unified tool to load them.


**Weaknesses:**

- For one specific blockchain(like Bitcoin), it doesn't cover all 6 tasks listed in the paper.
- Dashcoin does not have any labeled graph datasets and the task `Coin Tracking Across Blockchains` seems missing.
- The paper only presents the details and baselines of parts of the datasets, and I cannot find the details of other datasets on the website.


**Additional Feedback:**

No.


**Clarity:**

The paper is well written and provides enough background information for the reader.


**Correctness:**

Looks good.


**Documentation:**

The documentation looks enough to load the data.


**Ethics:**

The authors mention that they used the label cloud of etherscan.io to curate Ethereum address labels. It may violate the TOS/AUP of etherscan since the authors scrape the data from the company and distribute it. And the access to the etherscan labelcloud is not as public as the authors' claims in the paper, it needs an etherscan account to access the full data and may be restricted by the rate limit.


**Relation To Prior Work:**

Existing works and their limitations are discussed.


**Summary And Contributions:**

This paper makes a comprehensive dataset repository for blockchain graph research. It describes the limitations in existing solutions and provides a unified graph creation tool. This paper describes 6 different blockchain graph ML tasks and provides some datasets on Bitcoin, Ethereum, and Dashcoin. This work preprocesses the raw data and prepares the datasets from it, and the ML researchers who do not have enough hardware resources to preprocess the raw data can benefit from this work.

---

> ### Author Response · Authors · 2022-08-23
> **Response to Reviewer 4zbJ**
>
> We are grateful to the reviewer for the detailed and constructive questions, particularly on Dash-
> coin, and also for appreciating the novelty and potential of Chartalist.
>
> 1. For one specific blockchain(like Bitcoin), it doesn't cover all six tasks listed in the paper.
>
> In the paper, we summarize six different data analysis tasks over the blockchain networks. Some of these tasks are network specific since we do not have the same type of labeled data for all blockchains. Therefore, we could not cover all the tasks on all the blockchains. Still using different blockchains, our dataset provides benchmarks for six different tasks. As an example, the stablecoin analysis is achievable over Ethereum network. We presented all the available tasks based on the network type on our website, and the datasets can be downloaded separately for each task.
>
> 2. Dashcoin does not have any labeled graph datasets, and the task Coin Tracking Across Blockchains seems missing.
>
> For Dashcoin, currently, we provide the transaction network graphs. Dashcoin dataset can be considered as another real-world graph where the runtime performance of anomaly detection and graph-based ML learning algorithms could be tested.
>
> At this time, coin tracking across blockchains tasks does not have enough publicly available data for sharing. Hence the task is is not included, however, we are working to prepare labeled data for this task.
>
> 3. The paper only presents the details and baselines of parts of the datasets, and I cannot find the details of other datasets on the website.
>
>  Thank you for pointing this out. We have updated the website (chartalist.org) to contain descriptions of all datasets in terms of the number of data points or size, number of attributes, etc. We have also added and improved the visibility of links from the tasks to the datasets, and from the datasets to the tasks. For each task, we have also added a description of Baseline Methods to the website. Section 3.3 in the paper now also reflects the fact that we aim to continuously expand the number of blockchain datasets that Chartalist provides. This will allow for comparative studies. For example, the Dashcoin dataset can also be used to compare the results of models and algorithms devised for the Bitcoin dataset, as the underlying data structure is equivalent, but the data is smaller.

---

### Official Review · Reviewer_JJtm · 2022-07-28
**Interest dataset for graph ML in blockchain.**

**Rating:** 7
**Confidence:** 3
**Correctness:** Yes
**Clarity:** Yes

**Strengths:**

The paper is well motivated and easy to understand.

The benchmark provides exposure to blockchain and reduces the barrier to entry in the field. The application presented (wash trade detection, address classification, etc.) sounds really interesting.

There is a commitment to maintain and keep it updated.

The benchmark has dynamic multiplayer graphs with ground truth for anomaly detection.


**Weaknesses:**

**Cons/comments/suggestions/questions**

Authors should also include some GNN-based baselines in the work.

Following up on the above IMHO the comparative experiments could be more rigorous.

IIUC the wash trading task is constructed based on a certain heuristic. It may so happen the downstream ML model may learn just the heuristic without learning about the wash trading account distribution. I understand that it is not straightforward to get the ground truth for this. I think the implications of this require some discussion in the paper.


**Additional Feedback:**

See the weakness section.

**Documentation:**

Yes

**Ethics:**

- Contain any personally identifiable information or sensitive personally identifiable information: The authors claim that this dataset doesn't contain the PII directly but there is a chance that it can privacy implications. "Our data does not contain any PII. However, it is possible that if such PII were to be obtained from other sources, the datasets we are proposing here could lead to further privacy-related implications."

- Deceive people in ways that cause harm: Authors discuss that this dataset can be used by attackers to develop better attaching strategies.

**Relation To Prior Work:**

Yes

**Summary And Contributions:**

The paper proposes a labeled blockchain graph dataset from unspent transaction output and account-based blockchains called Chartalist. The dataset can be used for modeling, analysis, and representation of blockchain that can be used for applications like ransomware payment tracking, price manipulation, and money laundering detection. Chartalist provides transactions and blockchain extraction and graph creation tools for graph-based machine learning methods on the blockchain. The paper proposes 6 ML tasks and does a very good job of motivating and explaining the tasks.

---

> ### Author Response · Authors · 2022-08-23
> **Response to Reviewer JJtm**
>
> We would like to thank the reviewer for the detailed and constructive feedback, particularly on other DL tools and wash trading, and also for recognizing the novelty and potential of Chartalist.
> 1. Authors should also include some GNN-based baselines in work.
>
> Thank you for raising this question as indeed blockchain data offer a
> wealth of opportunities for GNN benchmarking. We have performed a broad range
> of experiments with GNN baselines on Ethereum tokens papers [Chen et al., 2021]
> and [Chen et al., 2022]. (For the sake of space, these results are not included in the
> manuscript but are presented on the website.) However, in the case of ransomware
> detection on Bitcoin, the current dataset has too few labels (i.e., less than 30K) to
> run GNNs; this is why we opt for simpler ML models there.
>
> 2. Following up on the above IMHO the comparative experiments could be more rigorous.
>
>  We agree with this concern, but we would like to reemphasize that the
> primary goal of this paper is to introduce the ML audience to the current key
> problems in blockchain analytics which remain largely uncharted territory for ML
> scientists and to provide easy one-stop-shop access to the blockchain data. For
> unlabelled Ethereum data, the rigorous experiments on GNNs and other DL have
> already appeared in ICML, ICLR. In turn, for labeled data, the experimental re-
> sults have appeared in IJCAI, WebConf, and KDD. Due to the space limitations,
> we cannot show all these results in the current manuscript, which also has a some-
> what different goal. In each dataset’s article, we compare our results to various
> models, such as the three heuristics in Bitcoin or known core decomposition tasks
> in Ethereum. That is, we already have extensive comparisons for each dataset. We
> hope that Chartalist datasets will be used extensively and new SOTA results will
> be achieved. We plan to prepare a leaderboard with SOTA results in the future.
>
> 3. IIUC the wash trading task is constructed based on a certain heuristic. It may so happen the
> downstream ML model may learn just the heuristic without learning about the wash trading
> account distribution. I understand that it is not straightforward to get the ground truth for this.
> I think the implications of this require some discussion in the paper.
>
>  Thank you for the comment. Indeed, an ML-based approach might come
> up with the same or a very similar heuristic. To the best of our knowledge, there
> does not exist any ground truth on wash trades on DEX. The primary implication is
> that the set of wash trades that an ML model finds should be checked for the aspect
> of no position change according to the legal definition – to absolutely make sure a
> set of trades can be considered wash trading. We have extended the description of
> 4.3 in the Experimental Setup and Baseline paragraphs to reflect this aspect.

---

### Review · Ethics_Reviewer_eC5r · 2022-08-22

**Recommendation:** 1

**Ethics Review:**

The reviewers have identified a few ethical concerns:

First, although the dataset does not itself contain personally-identifiable information, the authors mention that this information can be obtained from "other sources" and lead to "privacy-related implications," and that this is impossible to prevent "because all raw data is already accessible to users controlling significant hardware
resources." The authors should clarify in more detail what these other sources are and what these implications are, particularly as the audience at NeurIPS may not be familiar with blockchain privacy mechanisms. They give the example of IP tracking to connect transaction data to real-life identities, and assert that they do not have or provide this information in the dataset. Are there other concerns that should be mentioned here?

Second, a reviewer mentions that the authors' work might not be in compliance with the terms of service of etherscan. The authors should clarify how their work addresses this concern.

Third, a reviewer mentions that the dataset can be used maliciously, which the authors acknowledge. How might the authors mitigate this possibility? The authors may consider adding a registry (i.e., require the user to provide contact info or their institutional affiliation) to track downloads of the dataset.

---

### Meta-Review · Area_Chair_PQec · 2022-09-09

**Recommendation:** Accept
**Confidence:** 2

**Metareview:**

All reviewers acknowledge the quality of the paper and recommend acceptance, with scores ranging from 6 to 8. The authors have constructively engaged into the discussion with the reviewers, which has contributed to improving the evaluation of some reviewers.

The proposed dataset can have a significant contribution in attracting a wide community of scientists to analyze blockchain data. The datasets is, according to the reviewers, easy to manipulate and has proper documentation.

Ethical considerations were raised by the reviewers and the ethics committee. The authors have answered to these and, in our opinion, have brought the necessary clarifications to remove the concerns.

For these reasons, we recommend acceptance of the paper.

---

### Decision · Program_Chairs · 2022-09-16

Accept